**www.cambridge.org/ext**

## Overview Review

anthropogenic impact; conservation; ecosystem function; human dominance; keystone specieslate quaternary; megafauna; modern humans; worldview

**Corresponding author:**
Andrea Cardini;
Email: andrea.cardini@uwa.edu.au

# Destructive by nature? What human-driven extinctions of mammoths and mastodons mean for today's planetary environmental crisis

Andrea Cardini[1,2] 

[1]Dipartimento di Scienze Chimiche e Geologiche, Università di Modena e Reggio Emilia, Modena, Italy and [2]School of Anatomy, Physiology and Human Biology, The University of Western Australia, Crawley, WA, Australia

## Abstract

Scientists still debate whether small groups of Paleolithic hunter-gatherers caused the extinction of large Ice Age animals like prehistoric elephants, giant sloths and cave lions. Beyond paleontology, this question has deep sociological implications and is relevant for how we understand the role of humankind in today's environmental crisis. A human-driven megafauna extinction has often fostered the idea of a naturalization of human environmental impacts and the belief that all people (modern or ancient, rich or poor, from any part of the world) share responsibility for the current crisis. But is that true? In the review, I discuss whether a long evolutionary history of impacts really makes us inevitably destructive, compelling humanity to accept a devastating anthropocentric dominance as the fateful destiny natural selection built for us. In contrast, I argue that, while our exceptional ability to shape environments has made us a 'hyper-keystone' species, benefiting only a few species and humans, this same ability also has the potential to help us restore balance to the world. That requires rejecting anthropocentric supremacy and placing ecosystems at the center stage of our relationship with nonhuman nature. We may have wiped out the mammoths and mastodons, but human destructiveness is not fate.

## Impact statement

Mammoths, woolly rhinoceros, cave lions, thunder-birds, giant species of sloths, lizards and tortoises, as well as many other large terrestrial animals, including our own sister species – the Neanderthal – disappeared forever toward the end of the Ice Age. This is the largest extinction we know to specifically wipe out species of animals as large as or larger than an adult human (the so-called 'megafauna'). The role of modern humans in this event is still debated in the sciences and humanities. Indeed, to many scholars, a human-caused extinction of the majority of the most charismatic Ice Age land vertebrates seems crucial for shaping our response to the current global environmental crisis. If true, they argue, we cannot blame modernity for our devastating impacts, as human destructiveness starts in the Stone Age and is part of what we are. Thus, a significant role of a small population of 'low-tech' hunter-gatherers in the pre-agricultural megafauna collapse implies the inevitability of destructive human impacts and also equal responsibility for today's biodiversity crisis. In fact, this is a biased, deterministic view of 'human nature' that ignores scientific evidence and leads to the belief that modern humans can never coexist within a balanced natural ecosystem. In contrast, I maintain that human-driven global change reflects the interaction of biological and cultural evolution. Genetic change is slow, but culture is rapidly modified, and the direction of cultural change is not meant to be always negative for the ecosphere. Then, can we shift from harmful to beneficial impact by adopting a global ecosystem perspective? An objective interpretation of scientific evidence on human behavioral ecology suggests that the very abilities that make us a 'hyper-keystone' species are fundamental for restoring global ecosystem balance.

## Introduction

### *Tracing human supremacy from prehistoric extinctions to modern ecological and social domination: From megafauna to Native Peoples, 'ubi major, minor cessat'*

'One race of men — as in races of lower animals — shall disappear off the face of the earth and give place to another race … .The races of the Mammoths and Mastodons, and great Sloths, came and passed away: the Red Man of America is passing away' (General J. H. Carleton, commanding officer in the mid-18th-century US campaign against the Apache and Navajo, quoted by Ghosh [2021, p. 51]). As the Romans had it, '*Ubi major, minor cessat*': the weaker vanishes as someone

more powerful arrives. For General Carleton, Native Americans were, thus, destined to fade away, replaced by White invaders. Thousands of years before, as the Pleistocene (or 'Ice Age') was coming to an end and humans colonized the Americas, the largest animals of the New World met a similar fate of decline and extinction.

Today, few would accept genocide and ecocide as an inevitability of being humans. Yet, human supremacy and the domination of 'nature' are embedded in the leading worldview of our time (Crist, 2019). The presumption of human uniqueness creates an unsurpassable divide between us and the rest of nature. Not everybody shares this perspective, which, nonetheless, prevails in modern socio-political and economic systems (Ruuska et al., 2020). But what is the origin of human supremacism? Is it an existential cultural byproduct or is it an essential emergent trait of our biological evolution? In this context, can the extinction of mammoths and giant sloths be any relevant, beyond the disheartening similitude between the trajectory of population decline of both the largest animals of North America and its original inhabitants?

Perhaps, these are philosophical questions of little interest to biologists. Yet, when a student of my mammalogy class asked me where I stood in the debate that followed the first killing of a person by a brown bear in Italy, my instinctive reply was: 'Shall we live in a world where there is just one dominant species or do we want a balanced ecosystem enriched in its functions and diversity?'. Back then, however, I had not appreciated that the demise of native populations conquered by a technologically superior invader is just a special within-species case of a broader phenomenon of human ecological dominance (Ghosh, 2021). Thus, anthropogenic impacts on the environment do have potential sociological ramifications, which are also relevant for life and environmental scientists.

Carleton draws a daring (but not unusual in his time; Sayre, 2001) parallel between the massacre of Native Americans and the extinctions of the most charismatic species of the Ice Age, such as mammoths and other gigantic beasts, whose existence is now only known through fossil remains. In fact, alongside these giants, many other large (≥45 kg) land animals, the so-called 'megafauna', went extinct between 50,000 and 10,000 years ago (ya) (Koch and Barnosky, 2006; Stuart, 2015). The primary wave of extinctions, spanning ~40,000 years, mostly coincides with the Upper Paleolithic (~50,000–12,000 ya) and, thus, the end of the Pleistocene. However, this event is usually referred to as the Late Quaternary Extinctions (LQEs), because the extinction wave extended into the early Holocene (~12,000 ya to the present). The Holocene epoch, along with the Pleistocene, forms the quaternary period (~2.6 million ya to present), the shortest and most recent of the 'post-dinosaur' Cenozoic era beginning 66 million ya. It is largely because of the LQE that today's living terrestrial megafauna is downsized and impoverished (Malhi et al., 2016; Galetti et al., 2018). Thus, in the often heated discussion on the conflict that accompanies the recovery of the European populations of brown bears and wolves (Gross, 2023a), we seem to forget that these species are the only two survivors among all the large and diverse carnivores that once roamed this continent.

Carleton's parallel must imply humans as the cause of the megafauna extinctions to suggest that human dominance is key to both the destruction of native populations and ecosystems. Seen from the angle of the global environmental crisis of our time, that would also indicate a continuity between prehistoric extinctions of keystone megafaunal species and later anthropogenic impacts. It is precisely this continuity that a current of sociological research finds rich in potential societal and ecological ramifications. However, these implications are often overlooked in what we, as biologists, teach about conservation and environmental change, even though they are central to the debate on how we should respond to the current crisis.

## Magnitude and impact of LQEs on biodiversity and ecosystem resilience

During the LQE, 90 megafauna genera and almost 180 of the largest species of land vertebrates (≥45 kg) disappeared forever from the fossil record (Koch and Barnosky, 2006; Barnosky, 2008; Stuart, 2015). The event is global. It happens over a geologically brief time span of a few tens of thousands of years on all continents, except Antarctica. Nonetheless, there are significant differences in patterns, including the precise timing and severity (Koch and Barnosky, 2006; Stuart, 2015). The main commonality is the specific body mass range of the extinct species, with a strong prevalence of large and very large animals. Many weighed hundreds of kilos. Some (for instance, several species of elephantids, South American giant ground sloths and glyptodonts, or the Australian *Diprotodon*, the largest marsupial ever known) were colossal, exceeding one or more tons in weight. Using Europe as an example, where extinctions were more severe than in Africa but less than in the Americas or Australasia (Stuart, 2015), all terrestrial vertebrates heavier than 1 ton and, overall, more than ¼ of megafauna species, including, among the others, rhinoceros, elephants and mammoths, a giant deer, at least one species of horse, cave lions and cave bears, as well as our own sister lineage, the Neanderthals, vanish at a rate at least 10 times faster than in the previous 7,00,000 years between 50,000 and 10,000 ya (Stuart, 1999). In some cases, like the common hippopotamus, the lion and the spotted hyena, species go locally extinct, leaving behind an impoverished fauna, but they survive in a restricted area of their original range.

LQE often hit keystone species and ecological engineers, whose loss may have triggered cascades of extinctions (Malhi et al., 2016; Pires, 2024). Present-day large predators can have top-down regulatory effects on ecological networks (Ripple et al., 2016; Wilting et al., 2025). Their presence or absence, or even just a sharp change in population density, can trigger ecosystem shifts that radically alter species composition and the biomass and productivity of an ecological community (Ripple et al., 2014, 2016). A textbook example of this type of ecological interaction today is the Pacific sea otter (Estes, 2020). Its population density determines whether coastal regions remain in a highly productive, biodiverse kelp forest state, or, when otters are decimated by hunters, predators or disease, they shift to an impoverished sea urchin barren. As reviewed by Malhi et al. (2016), the largest mega-herbivores also play keystone ecological roles. Elephants promote long-distance large-seed dispersal (reducing inbreeding and allowing plants to track climate change and colonize new regions (Pires, 2024) and engineer the environment by creating open habitats, as they reduce woody species cover. A single African elephant can uproot hundreds of adult trees each year, but these animals may also promote the growth of woody vegetation by reducing competition with grasses and by moving nutrients inland from river floodplains, as well as by removing dry biomass and, thus, decreasing the frequency of fires in the savanna. Likewise, in wetter and colder climates, mammoths, horses, woolly rhinos and bisons facilitated the persistence of a steppe habitat at high latitudes. In this case, grasses might have increased transpiration in summer, avoiding upper-soil waterlogging and, therefore, reducing the occurrence of a, nowadays dominant, low-diversity mossy tundra (Malhi et al., 2016; Pires, 2024).

Although the population density of the biggest mega-herbivores is likely primarily controlled by bottom-up factors, such as primary productivity, the largest carnivore species may also contribute through predation on their young (Malhi et al., 2016). Conversely, if the largest predators significantly preyed on mega-herbivores, the population size of the herbivores could have exerted a bottom-up regulatory influence on the predators (Pires, 2024). Furthermore, larger predators directly (by killing) or indirectly (*via* competition) contribute to regulating the populations and food preferences of smaller megafauna carnivores and meso-predators. As they went extinct, this regulatory effect was lost, and the surviving predators expanded their ecological niche (Pires, 2024). Likewise, with fewer predators and without competition from other mega-herbivores, such as, for instance, mammoths and horses, the remaining species of grazers and browsers could increase population density and enlarge their distributional range, as it happened with the bison in North America (Pires, 2024). Thus, the megafauna ecosystem is characterized by mutual interactions between several keystone actors, such as the largest primary consumers and predators. These interactions are part of a much larger trophic network, whose complexity is still poorly understood and includes, among others, scavengers and an often overlooked biotic component, that of the megafauna-related microorganisms and parasites (Doughty et al., 2020; Pires, 2024). Clearly, the consequences of LQE went well beyond the loss of many of the most charismatic representatives of the terrestrial fauna (Smith et al., 2023). The local decline or complete disappearance of the majority of the Pleistocene mega-fauna had disproportionate effects on habitats and ecosystems, which have become impoverished and less functionally redundant, as the functions regulating fluxes of energy and nutrients now depend on fewer species. With loss of functions, or even simply because of a reduced number of animals performing those functions, ecosystems are less resilient and more vulnerable to further future biodiversity losses (Malhi et al., 2016; Pires, 2024).

Indeed, the LQE is the most selective event of the entire Cenozoic (our era, which begins 66 million ya) in terms of body mass of the extinct species, whose average size was between two and three orders of magnitude greater than that of surviving species (Pires, 2024). This selectivity alone implies downsizing and ecological downgrading of trophic networks, and, thus, a sharp reduction in landscape heterogeneity and functional diversity (Pires, 2024). As an extreme example, the megafauna of South America before the LQE consisted of species of herbivores that were on average larger than their African counterparts (Pires, 2024). After the LQE, none of the herbivores heavier than 1 ton survived in South America, and only tapirs and camelids are left among the species with body mass above 100 kg. Thus, as Pires (2024) remarks, today's Africa has an exceptional richness in very large herbivores, many of which perform important ecosystem functions, but that is mostly an artifact of the devastating effect of LQE in other continents. Earth, after the LQE and before the agricultural and industrial revolution, was already a very different place, one that, for the first time since life recovered from the mass extinction that wiped out all non-avian dinosaurs 66 million ya, was deprived of a diverse, abundant and ecologically functional megafauna (Malhi et al., 2016).

## From megafaunal collapse to planetary crisis: Is human destructiveness a natural trajectory?

LQE happens during the main wave of out-of-Africa dispersal of modern humans (*Homo sapiens sapiens*). While anthropogenic impacts seem like a likely contributor to the LQE, high-quality data on the subject remain scarce. Archaeological sites associated with

megafauna remains are, indeed, rare (Meltzer, 2015; Nagaoka et al., 2018) and it is often hard to precisely link the arrival of hunter-gatherers with specific extinctions. The Late Quaternary is also a time of rapid changes in climate and habitats, as the last glacial peaks and then ends. Koch and Barnosky (2006) and Stuart (2015) debate hypotheses and review the evidence, but new studies have been published since then. For instance, two very recent examples, with extensive examinations and sophisticated modeling of the most likely factors, are Stewart et al. (2021) and Lemoine et al. (2023), who reach opposite conclusions, one blaming the climate and the other humans.

Whether human hunters alone (as in the original 'overkill hypothesis' (Martin, 1966), climate change, other factors or their complex interaction (Koch and Barnosky, 2006; Stuart, 2015), have been responsible for the LQE is not the focus of this essay. Having stressed the large uncertainties on causes, I will assume a decisive role of humans in order to discuss the potential implications of the LQE for the modern global environmental crisis. For this assumption to hold, one does not need to postulate a rapid overkill of large animals by humans, as foragers rapidly spread across continents where prey had not evolved effective anti-predator responses to skilled social hunters with relatively sophisticated weapons, including the use of fire. The role of humans is 'decisive' even if it was their interaction with rapid changes in the climate, and, thus, with habitat modification, that caused the extinctions. In that case, humans would share the blame with geological forces, but, without us tipping the balance, the Ice Age megafauna would have largely survived environmental change as most of its representatives did in previous glacials and interglacials.

What, then, are the implications of a potentially large-scale extinction event caused by a relatively small number of hunter-gatherers, whose global population at its peak may have been half that of present-day London (Hawks et al., 2000; Eller et al., 2009), as they emerged from their African homeland toward the end of the Ice Age? The vast majority of LQE occur before there is any radical change in human behavior and ecology (Scerri and Will, 2023). Gradual innovations, better tools and, likely, a progressive development of more complex forms of symbolic communication were facilitated by the demographic expansion of the Upper Paleolithic, but 'modern human behavior had deep African roots' (Scerri and Will, 2023, p. 13). Thus, if vast anthropogenic extinctions predate the modern lifestyle of the agricultural and, later, industrial societies, it becomes harder to point at these specific cultural revolutions (O'Brien and Laland, 2012) as the root causes of human environmental destructiveness. Nature came before nurture.

If destructiveness is the essence of our species, this begs an important question: 'Is the human impact natural?' (Crist, 2019, p. 83). In an investigation of the deeper reasons for our current environmental problems (Nelson and Lemberger, 2021), Crist (2019) devotes an entire chapter to this conundrum. The interpretation of the LQE plays a central role in her argument (Crist, 2019, pp. 102–104): the LQE '… [often] attributed … to the global spread of human hunters … [seems to invite] an irresistible inference … that the anthropogenic extinction crisis is a single continuous event beginning with the human diaspora out of Africa … and continuing all the way to this morning'. Thus, she adds (p. 108–109), 'the prevalent belief that human nature is to blame for the ecological crisis offers a seductively simple explanation that discourages deeper thinking … naturalizing the human impact on the biosphere … because humans are essentially different from the rest of life'. However, in sharp contrast to this view, her definitive conclusion (p. 109) is that 'human supremacy is a social achievement … not a biological inheritance'.

The excerpt from Crist's book provides a useful summary of the seemingly obvious inference of an anthropogenic origin for the LQE, but also of a less apparent, more unsettling, implication: the naturalization of human devastating impacts. Like several other scholars, particularly in the humanities and social sciences (*e.g.*, Deloria [1997] cited by Sayre [2001]; Robins [2024]), Crist rejects both the idea of an unlikely continuous pattern of extinctions over the past 50,000 years and, even more strongly, the tendency to naturalize and, therefore, normalize, human dominance over Earth's ecosystems, which she attributes instead to cultural factors. Similar points have, actually, been made also by natural scientists. For instance, Louys et al. (2021, p. 1), in their examination of archaeological and paleontological evidence from Pleistocene islands inhabited by hominins to assess 'whether humans have always been destructive agents', concluded that 'It is not until the Holocene that large-scale changes in technology, dispersal, demography and human behavior visibly affect island ecosystems. The extinction acceleration we are currently experiencing is thus not inherent but rather part of a more recent cultural complex.'

Since I have assumed, for this discussion, that the LQEs were driven by our species, the implication, however, appears inevitable. Human environmental destructiveness is not an existential aspect of the 'modern lifestyle' ('a social achievement'). If this were the case, its emergence should come after, not before, the cultural revolution marked by the progressive domestication of plants and animals, which began during the early millennia of the Holocene and became a global phenomenon between 8,000 and 5,000 ya (Ellis et al., 2013; Taiz, 2013; Stephens et al., 2019). Farming dramatically transformed us and the land (Ellis et al., 2013; Taiz, 2013). However, if anthropogenic impacts had been devastating the planet well before agriculture, then, a massive ecological footprint becomes a defining trait of human evolution. Thus, not even Paleolithic hunter-gatherers were in harmony with the ecological community they belonged to. As these foragers colonized all nonfrozen continents, they tipped the balance of ecosystems that had, for the most, robustly survived hundreds of thousands of years of dramatic changes in the climate and environment. In this scenario, humans are seen as a consistently hyper-dominant member of the ecological communities for no less than 50,000 years. To put it bluntly, in the light of the consequences of this dramatic dominance, we, as a species, can indeed be seen as 'destructive by nature'.

Following this line of reasoning, dominant anthropogenic impacts become part of our natural history ('a biological inheritance' in Crist's terms). In the sociological interpretation, an essentialist conclusion becomes an evolutionary fact: 'This form of species narrative allows people to assume that ecologically destructive behaviors arise from their genetic humanity and not from the cultural and political structures' (Robins, 2024, p. 18). But is it true?

Of course, there are no genes for environmental destructiveness. Natural selection, throughout the history of our lineage, has led to the coevolution of a combination of traits, such as intelligence and sophisticated tool use (including the control of fire), a complex syntactical language that facilitates cooperation and cumulative advances in culture and so on (Richerson et al., 2021). This combination has predisposed our species to an exponential increase in its ability to colonize and successfully survive in almost any type of habitat. Progressively, we also discovered new resources to generate energy and labor, such as domestic species during the Agricultural Revolution and fossil fuels during the Industrial Revolution (Fischer-Kowalski et al., 2014). Culture, here defined as the social transmission of information (Bridges and Chittka, 2019), was key to progress. Behavioral innovation, tool use and the capacity of rapidly

learning from other individuals and groups, and to cooperate in larger numbers (Boyd and Richerson, 2024), allowed to flexibly reshape the ecological niche of humans in relation to the specific conditions (Henrich and McElreath, 2003). This, in turn, might have led to the preferential selection of gene variants that are favorable in a human-modified environment and, therefore, to a runaway process of accelerated gene-culture coevolution (Laland et al., 2010; Rendell et al., 2011; O'Brien and Laland, 2012). As Rendell et al. (2011, p. 833) wrote, 'effective niche-constructing capabilities generate selection for more potent niche-constructing capabilities, in a self-reinforcing dynamic'. For instance, the domestication of animals for meat and dairy farming might have brought a selective advantage to herders and, at the same time, if being able to drink milk after weaning increased fitness, created an environment where lactose tolerance in adults could evolve (Laland et al., 2010). Rapid gene-culture coevolution likely played a central role in transforming us into a hyper-keystone species that has a disproportionate impact on ecosystems and affects multiple other keystone species (Worm and Paine, 2016). Usually, our impact is harmful to biodiversity and ecological equilibrium, and it frequently happens on a planetary scale. Thus, as a 'side effect' of our own success, we began exhausting resources, leading other life forms to extinction and shifting ecosystems toward degraded states and a precarious balance. Yet, despite local failures, and even occasional societal collapses (Scheffer, 2016), in the evolutionary short term of a few tens of thousands of years, humanity has managed to escape the consequences of environmental damage, increasing its population from a few million Paleolithic foragers to the present-day 8 billion. However, that the process, as powerful as it is, cannot be sustained indefinitely is clear in Rendell et al.'s words (p. 833): 'Nonetheless, most resources cannot rise without limit, so at some point an upper bound will be reached, and the opportunities for the evolution of more potent niche construction will diminish'.

If it is a fact that humans are 'destructive by nature' (a first implication of the 'irresistible inference' of a continuity in anthropic impacts), we have not just discovered an accidental, but rather unfortunate, outcome of our evolutionary history. For the social scientists, naturalizing destructiveness also leads to further and profoundly disturbing corollary claims that can bias, if not undermine, how contemporary society responds to the planetary environmental crisis we face. To begin, since the emergence of modern humans in evolutionary history, true wilderness has never existed. If no human-inhabited environment is, or ever was, pristine, it follows that we all, in every region of the world, regardless of historical socioeconomic and demographic differences, share the responsibility for the anthropogenic devastation of the world's ecosystems. Besides, even if we are not (and cannot be, from an evolutionary and ecological perspective) separate from nature, a massive ecological footprint defining the human species and its interaction with the environment suggests inevitability: this is part of what we are as a biological species. At best, as humanity confronts the threats of global environmental change, impacts might be mitigated and, if achievable, their directions managed to benefit first and foremost our own species. Thus, any intervention is bound to happen with us, *H. sapiens sapiens*, as the center stage, as that was and remains the niche our species has evolved to fill.

## Change and responsibility: Who bears the burden for the twenty-first-century environmental crisis?

Are we, all living people of all continents and all those who came before us, equally responsible for the anthropogenic environmental

crisis of the twenty-first century? The answer to this question is the easiest: we are not, even assuming that Paleolithic hunter-gatherers drove the Ice Age megafauna to extinction, dramatically altering the ecosystem balance on land (Malhi et al., 2016; Pires, 2024). We might be 'destructive by nature', in the sense I outlined before. However, there are differences in quality (debatable) and quantity (indisputable).

A few words about quality, first. A human-driven LQE event is a contingent accident. Paleolithic foragers are most unlikely to have intentionally driven any species to extinction. This is in sharp contrast with, for instance, the systematic and meticulous persecution of the thylacine in Tasmania by white farmers, coupled with the destruction of its habitat, until the wild population declined and the last captive individual died (Sleightholme and Campbell, 2016). Likewise, it is also radically different from the large-scale and deliberate eradication of wolves by European settlers and their descendants in most of North America (Musiani and Paquet, 2004).

Besides, we cannot judge indigenous populations that might have eroded biodiversity and degraded ecosystems in the past using contemporary understanding and values (Taylor et al., 2016). There is a difference in knowledge, intentionality and awareness. In Best's words (Best, 2021, p. 9), 'extinctions did not always arise from the tips of human spears and orgies of killing but rather one species, among others, exploiting an ecological niche for survival, perhaps unaware of species limits and finitude and the consequences of its actions'. Today, in contrast, a large proportion of humanity is well aware of being the driver of a massive demographic decline in wildlife populations and a myriad of very recent anthropogenic extinctions (Ceballos et al., 2017; Cowie et al., 2022). Likewise, pollution and deforestation, just two of many possible examples of environmental degradation, make daily headlines in the media, and we are increasingly learning about the loss of ecosystem functions and the dramatic shift in ecosystem states accompanying the disappearance of keystone species (Malhi et al., 2016). Yet, aware as we are, we seem unable to stop.

One might argue that, tens of thousands of years ago, as today, ours is the only species able to drive other life forms to extinction on a geological scale (Ceballos et al., 2017; Cowie et al., 2022). Wherever we have been, we seem to have left evidence of our devastating ecological footprint (Denevan, 1992; Foley et al., 2013; Ellis et al., 2021). There might be a few exceptions, if any. According to a Cherokee scholar, even 'modern Natives and their ancestors are neither saints nor sinners in environmental matters – sometimes they have degraded their environments even to the point of societal collapse' (Weaver [1996, p. 7] cited by Taylor et al. [2016]). However, whether or not one agrees that the core of who we are and how we interact with other species and the environment has changed little over the past 50,000 years, the quantitative differences in impacts between then and now are undeniable. These are measurable. A few examples should suffice to remove doubts on magnitude and responsibility. Let us focus primarily on assessing biodiversity and environmental impacts before and after the Industrial Revolution, a period for which we have uncontested evidence. If it can be demonstrated that, despite a long history of widespread extinctions and global impacts, the decline of wild species and habitat degradation was relatively modest before the industrial era, the assignment of responsibility becomes clear: the blame lies with the dominant industrialized nations.

The majority of scholars acknowledge the exponential escalation in anthropogenic impacts in the last few centuries, and a further acceleration after World War II (Crutzen, 2006; Foley et al., 2013; Smith and Zeder, 2013; Steffen et al., 2015; Lewis and Maslin, 2020).

Crist (2019) herself writes (p. 107) that 'humanity's blow on the biosphere has increased by orders of magnitude in comparison to what occurred previously in the Holocene'. Accordingly, Crutzen (2006, p. 14) offers a number of examples to support his proposal of a new geological epoch, the Anthropocene, marked by a massive planetary increase in anthropic influence: 'During the past … centuries human population increased tenfold to 6000 million, growing by a factor of four during the past century alone … accompanied for example, by a growth in the cattle population to 1400 million … Similarly … industrial output … grew forty times. More than half of all accessible fresh water is used by humankind. Fisheries remove more than 25% of the primary production of the oceans in the upwelling regions and 35% in the temperate continental shelf regions'. Now, two decades after Crutzen's proposal, an even larger population and an additional increase in global consumerism have further magnified those impacts (Fletcher et al., 2024). For instance, $CO_2$ is more than 50% higher than in preindustrial time and still on the rise (Friedlingstein et al., 2023), but climate change is just one of many critical aspects of the present global environmental crisis: six (biosphere integrity, changes in land and freshwater systems, alteration of biogeochemical cycles of nutrients, etc.) of nine planetary boundaries for a safe operating space for humanity have been crossed (Richardson et al., 2023) and we rapidly approximate tipping points of irreversible change (Lenton et al., 2019; Tollefson, 2023).

That the Industrial Revolution has been a multiplier of the ecological footprint of humanity, as never before, can be appreciated also by measuring the massive global reduction in wildlands, the steep increase in the rate of extinction and the different magnitude of the sociometabolism of foragers, farmers and industrialized societies. Using a definition of wildlands as uninhabited regions under no intense use, Ellis et al. (2010) conclude (p. 589) that 'in 1700, nearly half of the terrestrial biosphere was wild … [and] most of the remainder was in a seminatural state (45%) having only minor use for agriculture and settlements. By 2000, the opposite was true, with the majority of the biosphere in agricultural and settled anthromes [*i.e.*, anthropogenic biomes], less than 20% seminatural and only a quarter left wild'. Wilderness has, thus, been vanishing before our eyes in the last 250 years. 'Wild' does not mean 'untouched'. Prehistoric activities by humans might have reshaped habitats and ecological communities, leaving a long-term legacy on the biosphere (Ellis et al., 2021). However, the strongest impacts, responsible for the current crisis, are a consequence of the direct or indirect appropriation of land and the intensification in the use of resources by industrialized societies (Ellis et al., 2010, 2013, 2021). Likewise, for millennia we have been responsible for species extinctions on islands and continents (Johnson et al., 2017). Yet, it is in the last three centuries that the rate of vertebrate extinctions has risen to no less than 100 times higher than the background rate estimated using the fossil record (Ceballos et al., 2015). This looks impressive, but population declines and species loss are underestimated, as we lack accurate information on the past, and even today, only a minority of taxa is carefully assessed by conservationists (Hughes et al., 2021). Thus, it might not be unreasonable that, since 1,500 AD, 8–13% of the currently known 2 million species have been lost (Cowie et al., 2022) and, as in land vertebrates (Ceballos et al., 2015), this has likely happened with an accelerated pace in the last few 100 years.

In the attribution of the responsibility of anthropic impacts, however, the type of economy and the prevalent societal lifestyle (a term here used, for brevity, to broadly refer to how people live their lives (Agnew et al., 2023) matter. In calculating the human

ecological footprint (Wackernagel et al., 1999; Toth and Szigeti, 2016), consumption is a key factor and also, often, the largest contributor to species extinction risk (Irwin et al., 2022). For instance, the overconsumption of meat products in the European Union (EU) is linked to the destruction of the Amazon forest and its biodiversity. Estimates (Rajão et al., 2020) suggest that ~20% of Brazilian soy, mostly imported in Europe as cattle feed, and almost 50% of beef from Brazil, which provides up to 40% of EU imports, come from illegally deforested areas in this country. The destruction caused to tropical forests around the world by consumption in Europe is such that it has been argued that the 'Europe's Green Deal offshores environmental damage to other nations' (Fuchs et al., 2020, p. 671).

Measuring societal consumption is, thus, important to appreciate direct and indirect anthropic impacts and, therefore, differences in responsibility. A useful concept to convey the magnitude of variation in the consumption of energy and goods during the major transitions from hunter-gatherers to farmers and from the latter to the modern industrialized economy is the metabolic rate of a society or its 'sociometabolism'. The sociometabolism represents 'the rate at which energy is exchanged between a human society and its environment, and transformed within a society' (Malhi, 2014, p. 144). With this type of measure, we can estimate the overall material and energy requirement of a person within a given type of society, as if the society were a 'super-organism' with a metabolism. Thus, expressed in kcal per day, an average hunter-gatherer needs ~6,000 kcal/day, which is three times the 2,000 kcal/day physiological basal metabolism of a person at rest (Malhi, 2014). In contrast, a European preindustrial farmer used about 42,000 kcal/day, and a citizen of a highly developed nation, such as the United States, consumes almost 250,000 kcal/day (Malhi, 2014). This means that, in terms of natural resource use, the average American uses as much as 40 hunter-gatherers or six preindustrial European farmers, with each of those farmers using as much energy as seven Paleolithic foragers. Thus, Fischer-Kowalski et al. (2014, p. 8) observe that 'until … [around AD 1500], human population growth and metabolic rates carry about equal weight in increasing human pressure on the environment approximately fivefold from the year AD 1 onwards. From then on, the overall pressure of humanity upon the Earth increases by one order of magnitude; energy intensity contributes to this rise by roughly tripling the impact of population growth. Technology, because it is based upon a shift from biomass to fossil fuels … does not moderate this impact, but enhances it by a factor of 1.5'. They are also clear (p. 25) that it is with the incipient Industrial Revolution that 'a veritable take-off can be observed. From 1700 onwards, human impact doubles every century, from 1900 on it doubles in 50 years, and from 1950 on it triples in 50 years, with no sign of saturation yet'. This clearly shows that the biggest rise in human impact on the environment, although already noticeable a couple of centuries before industrialization began, is recent, consistent with previously mentioned indicators such as extinction rates, population growth, $CO_2$ emissions and the amount of wildlands.

With developing countries having a sociometabolism 5–10 times lower than that of industrialized nations, we arrive at a less obvious conclusion: in modern times, the impact of the socioeconomic system far outweighs that of population growth (Krausmann et al., 2008). The direct implication is that societies that first transitioned to the industrial system have historically contributed the most to environmental damage – about 10 times more than farming societies and roughly 50 times more than hunter-gatherer societies (Krausmann et al., 2008; Fischer-Kowalski et al., 2014). These industrialized

nations continue to have a disproportionate ecological footprint, despite smaller populations, when compared to developing countries and less wealthy societies. Thus, going back to greenhouse gas emissions as an example of impact, the wealthiest 10% of the world population (mainly the ~700 million citizens of Europe and North America) was responsible in 2015 for 48% of the total annual emissions, while the 50% bottom income earners (the poorest ~3.7 billion people) emitted overall just 7% of those same atmospheric pollutants (Capstick et al., 2020). However, it is not just current emissions that are driven by rich, industrialized countries. Since 1800, 86% of the contribution to the total increase of $CO_2$ in the atmosphere has been attributed to highly developed countries (Neumayer, 2000). It seems, at a first glance, paradoxical that precisely by using the energy of fossil fuels we have managed to increase today's total mammal biomass of the land megafauna about seven times compared to its estimate before the time of the LQE (Barnosky, 2008). However, even including marine mammals, and thus the whales and the other giants among living animals, 96% of the total modern-day mammalian biomass is made of humans and their livestock, and just 4% is wild species (Bar-On et al., 2018). This massively unbalanced ratio indicates a similarly unbalanced ecosystem and is another feature of modernity, since for most of the Pleistocene, and even at the time of the major collapse of the Ice Age megafauna, the human biomass was a fraction of the total biomass of all land mammals (Barnosky, 2008).

Again and again, we have demonstrated that the ecological footprint of modernity is incomparably heavier than any damage caused in our hunter-gatherer past. Besides, since the spread of agriculture, impacts have been magnified multiple times, and later, with the Industrial Revolution, they have further increased by at least an order of magnitude. For this last and most substantial rise in anthropogenic pressure, the responsibility can be attributed for the vast majority to the highly industrialized Western nations that have so far received the greatest benefit (e.g., the proportion of contribution to the $CO_2$ concentration explains 2/3 of 1990 differences in per capita gross national product (Neumayer, 2000) to the detriment of the poorest and the environment. In terms of extinctions, the differential contribution is so remarkable that, by quantifying the consequences of consumption on biodiversity, researchers can show how virtually all rich industrialized European and North American countries are, in fact, importing the extinction risk (i.e., causing extinctions elsewhere) from the developing nations, whose resources they are exploiting (Irwin et al., 2022).

To summarize, human-driven LQE cannot be used to argue a shared responsibility of humanity through time for the global environmental damage and the unbalanced ecosystem we have created. In fact, even if we narrowly focus on modern impacts caused by industrialized societies, by far the largest of any time, both currently and historically, the responsibility mostly rests on the shoulders of the richest people and nations. Anthropogenic LQE, therefore, do not make all humans and socioeconomic systems equally impactful: as we may have to acknowledge an early origin of human 'destructiveness', we must also recognize that an ecological footprint so huge to alter the climate and cause a rate of extinctions comparable to that of the past five great mass extinctions of geological origin is an unprecedented emerging aspect of the last 250 years of Western-dominated industrialized economy. If anything, clarifying similarities and differences between human-driven extinction events and the related large-scale loss of ecosystemic balance contributes to understanding how the massive inequality within and among nations in the more recent history of humanity is at the base of the current environmental emergency

and, thus, a priority to address for any effective solution to the planetary crisis (Ross, 2017; Best, 2021; Fletcher et al., 2024).

## Rethinking the myth of human incompatibility with balanced ecosystems

Even without any 'genes for destructiveness', and even when differences in the magnitude and responsibility of our past and present impacts are acknowledged, one might still wonder whether an anthropogenic LQE ultimately implies a fixed trajectory of human behavior, one that leaves no alternative but to hopelessly document the ongoing destruction of life on Earth. Alternatively, as Nagaoka et al. (2018) argued, '… like any other organism, humans can destroy, modify, enhance, or preserve depending on context. And there is an extensive continuum of human–environment interactions that range from extinctions to sustainable coexistence' (Nagaoka et al., 2018, p. 9692). Thus, for these scholars, in their evaluation of how different disciplines interpret the overkill model, a major conclusion is that there is no inevitability in human destructiveness as long as we can profoundly reshape our niche in a direction different from the one prevailing since the Agricultural Revolution. Who is right?

In this last section, I will further explain why Nagaoka et al. are correct and argue that, regardless of the onset of widespread anthropogenic impacts, human nature offers no excuse for inaction in the face of an increasingly threatening global environmental crisis. Indeed, we may be 'destructive by nature' with a long chain of anthropogenic extinctions that begins 50,000 ya, if not earlier. However, as anticipated, the most profound changes for us and the planet are much more recent and mostly driven by culture. First, the domestication of animals and plants and, later, the industrialization of production and society demonstrate the enormous potential consequences of a culturally dominated niche construction in humans. Regrettably, these revolutions, as they led us to modern civilization and 'progress', have also created the unprecedented environmental crisis that threatens our own future survival (Wright, 2004). They transformed *H. sapiens sapiens* in a geological force of destruction that can change the face of the planet to meet its exclusive short-term needs (Ehrlich and Ehrlich, 2008). The root of the current crisis may well be that 'the will to dominate exists within us in seed form, affiliated perhaps with life's imperative for self-preservation' (Crist, 2019, p. 112) but it is 'the anthropocentric worldview' that has tilled and fertilized 'that particular seed, growing it into a … now global superstructure of domination'. Much like a positive feedback loop, cultural innovations led to progress, which increased human dominance. This, in turn, tends to reinforce anthropocentrism, promoting the idea of human supremacy and further domination. Over time, this dynamic spiraled out of control, reaching its apex with the ubiquitous Western socioeconomic system of the capitalist free market. To be fair, although neoliberal capitalism currently dominates with devastating impacts, communism and socialism had no better environmental record (Change, 2011).

Having acknowledged the overwhelming responsibility of industrialized nations in the modern crisis, one might also note that if no totally pristine habitat remains on Earth, and if anthropogenic impacts extend back to historical or even prehistoric times, then, despite the diversity of human behaviors and the differences in socioeconomic structures and belief systems, there appear to be few examples of human societies whose footprint on the environment and biodiversity was truly negligible (Foley et al., 2013; Boivin et al., 2016; Hussain and Baumann, 2024). Nonetheless, humans

have achieved an undeniably impressive demographic success, and on average, we also live longer and more comfortable lives. This reading of human history is, however, misleadingly myopic (Wright, 2004). Human success is measured on a minimalistic evolutionary timescale of centuries or millennia, despite looming preoccupation and uncertainty about our own future in the next decades (Gross, 2013; Best, 2021; Gross, 2023b). Furthermore, the benefits of 'progress' were and are highly unequally distributed, and originate from colonialism and economic imperialism (Ross, 2017; Lewis and Maslin, 2020; Ghosh, 2021; Hickel et al., 2022) that can be seen as an intraspecific variant of supremacism. A just system should have already eradicated poverty and provided all of humanity with its basic needs. Furthermore, innovations have nearly constantly led to a progress trap, where damage is either externalized (i.e., inflicted on those that did not cause it) or requires further innovation to be fixed (Wright, 2004; Ellis et al., 2013; Lewis and Maslin, 2020; Best, 2021). This is how, for instance, the slow progressing and often problematic (Bell, 2015; Sovacool, 2021; Gross, 2024) 'ecological transition' of wealthier industrialized nations outsources social and environmental damage to developing countries by exploiting their natural resources (Fuchs et al., 2020), as with Chilean copper or Congolese coltan and cobalt used in low emission electric vehicles (Sovacool et al., 2020). Likewise, the supraspecific application of the same supremacist philosophy is wiping out thousands of species, destroying habitats, polluting air, soil and water and degrading and downgrading ecosystems (Crist, 2019; Ghosh, 2021). Environmental impact interacts with social injustice, as it hits harder the poorest among people and nations, ironically, precisely those who contributed the least to the crisis (Chancel, 2022; Fletcher et al., 2024; Wilkinson and Pickett, 2024).

Transitioning to a radically new socioeconomic system is challenging and faces resistance (Büchs and Koch, 2019), but these difficulties do not originate from a potentially inherent destructiveness of humans. They derive from the structural quality and complexity of an ill-designed ecological niche constructed to achieve human dominance with no concern for ecosystem balance and other life forms (Büchs and Koch, 2019). Yet, the direction for change is clear: we are part of the Earth ecosystem, and a stable economy and thriving society depend on its long-term health and robustness (Ruuska et al., 2020; Dirzo et al., 2022; Fletcher et al., 2024). The United Nations Sustainable Development Goals (SDGs) have, so far, been largely contradictory and ineffective (Hickel, 2019; Cardini, 2024; Montesano Montessori and Lautensach, 2024). However, they do present, at least in theory, a realistic vision of priorities (Montesano Montessori and Lautensach, 2024), as illustrated by the SDG 'wedding cake' model (https://youtu.be/dcvz6Fv8DqU), where the biosphere forms the broad foundational layer supporting society, which in turn underpins the economy.

The ecological niche we have constructed rests on a no longer tolerable supremacist assumption of inequality. This niche is part of our phenotype, a complex manifestation of the interaction between our genes and the environment. However, it is a niche that, in *H. sapiens sapiens*, has been largely dominated by culture. Genetics may set the direction of our behavior, but the final outcome, what we end up doing, depends on the specific environment, which in humans is strongly shaped by culture. Thus, even the most fundamental instinct, without which there can be no evolution by natural selection, that is, the desire to survive and reproduce, is not destiny (Alcock, 2001). Many people choose to have few or no children at all. From an evolutionary standpoint, this is a contradiction unless it somehow benefits one's relatives, like in the concept of kin selection (Birch, 2019). While this can happen in some cases, it is

probably uncommon in modern societies. Surely, we did not choose either our genes or the environment (Sapolsky, 2021) that led us to this 'decision', but the mere fact that, given a specific context, the outcome goes against the strongest genetic predisposition, that of leaving a progeny, shows that genes, our 'essence', are not fate (p. 182 and following ones in Alcock [2001]).

Similarly, while our species may have become a hyper-keystone species (Worm and Paine, 2016) capable of dominating all other components of its ecosystem, this does not mean we must do so. On the contrary, acknowledging this potential can help us transform dominance into a balanced, ecosystemic coexistence, an essential step for ensuring our long-term evolutionary survival as part of a diverse and resilient ecological community (Washington et al., 2017). We are masters of niche construction and 'niche construction is frequently influenced by prior selection, but it is, at most, only partly determined by genes' and largely depends 'on environmental influences and on other inherited information and materials that together shape the developing organism and fashion how it interacts with the world' (O'Brien and Laland, 2012, pp. 436–437). Even if our natural history had made us uniquely able to change the planet with little oversight for the future, it did not remove us from the long-term reliance on the global ecosystem, whose functions go well beyond providing short-term services for humans (Washington et al., 2017). Indeed, our epoch 'ought to be characterized not by even further distending human hubris, but by humility and respect', as we 'cannot ignore the weight of the arguments that provide compelling reason to believe that at least some parts of nonhuman [life] have intrinsic value, and therefore deserve direct moral consideration' (Batavia and Nelson, 2017, p. 374).

Finally, when examining the potential implications of the specific pattern of human-driven extinctions through time for both the attribution of responsibility and the direction we choose to take in our relationships with the biotic and abiotic components of the world ecosystem, we should start from the same fundamental premise of sociobiological research (Segerstråle, 2000, p. 389): 'The problem is … the idea that our moral judgments ought somehow to be tied to the latest scientific knowledge [but]… it is exactly this connection between scientific understanding and perceived social utility that needs to be broken, not least for the reason that scientific knowledge and the interpretation of facts changes over time'. Thus, whether we truly are 'destructive by nature' and what precisely this means are elements of a complex and dynamic system of scientific investigation that constantly revises its results. This system provides useful information in the realm of ethics and behavior, but it offers neither a prescription nor a justification for how we should act. Human destructiveness may be the outcome of an evolutionary history that made us the indisputable champion of niche construction. Dominance and supremacism are problematic parts of the niche we built. As we created this ecologically distorted worldview, we can also learn to deconstruct these devastating aspects of our culture to rebalance the human niche within the planetary ecosystem. There is no essentialist obstacle to such a sharp change, and there might be a multiplicity of ways to shift from an anthropocentric to an ecosystem-centered direction. Whether in practice this is feasible rapidly enough to counter the impending environmental disaster is, however, a much harder question to answer.

The first modern humans who colonized the Americas might have caused extinctions and environmental change. That there, and in other continents, a tiny population of 'low-tech' foragers was potentially able to irreversibly and radically disrupt the balance of entire ecosystems, invites profound considerations on the sustainability of the human enterprise in the twenty-first century. To what extent LQE, if human-driven, was an accident or might have occurred with a degree of intentionality is hard to say. Most likely, Paleolithic hunter-gatherers were initially unaware of the consequences of their impact. Unlike them, in sharp contrast, but similarly to General Carleton, we know we are leading other life forms to extinction and can clearly see the environmental devastation we bring and the perils ahead of us. Tragically, we may lack a deep appreciation of the urgency and gravity of the crisis. Yet, the end of human dominance should be, for us, the beginning of the next ecocentric cultural revolution of *H. sapiens sapiens.*

**Open peer review.**   To view the open peer review materials for this article, please visit http://doi.org/10.1017/ext.2025.10007.

**Acknowledgements.**   The author is grateful to Eileen Crist for the brief discussion they had on the implications of the LQE, which contributed to the author's motivation to further explore an issue that the author has been thinking about for years and became even more interested in after reading her thought-provoking book *Abundant Earth.* The author would also like to thank the editors, as well as two anonymous reviewers, for their comments, which helped the author improve the article by better stressing its focus and implications. This article is dedicated to the memory of Maria Zaytsava (2001–2025) for her idealism and courage against injustice and the domination of the powerful.

**Author contribution.**   The author designed and wrote the whole study. AI-assisted technologies (ChatGPT 4) were used only to improve English. The author has reviewed and takes responsibility for the final text.

**Financial support.**   There was no financial support.

**Competing interests.**   The author declares none.

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
