## [Reviewer Report]

From the abstract, the manuscript appears to aim to challenge the notion that humans are inherently destructive and instead emphasize the capacity of humans to bring about positive change. However, the manuscript’s structure and argumentation do not clearly follow this path. Instead, the first half of the manuscript focuses on Late Quaternary extinctions (LQE) as evidence of human destructiveness, eventually taking the stance that this trait is a given. The manuscript then proceeds to describe modern environmental impacts associated with human actions, identifying mass consumption and exponential population growth as significant factors contributing to the current ecological situation. The conclusion appears to be that human “culture” offers a means to address the environmental degradation that is occurring today. However, the consequence of this argument structure is that the link between the supposed human-caused LQE extinctions and modern environmental degradation breaks down. This leaves the manuscript feeling like it is telling two separate stories.

The conceptual fingerprints of the overkill model are found throughout the manuscript. Unfortunately, the use of overkill in the argument automatically forces the discussion of human impacts to destructiveness as an inherent and fixed trait. However, the second half of the manuscript argues for the ability of humans, through culture, to overcome environmental challenges. In other words, the first half is based on the argument of the fixed nature of human behavior, while the second half leans into the idea that human behavior can strive for solutions. To be logically consistent, the manuscript could retain the LQE premise with its fixed human behavior and dedicate the second half to documenting modern destruction, with no solutions or hope. Alternatively, the manuscript could be reoriented to shift the prehistoric narrative to focus on human behavior as situationally diverse and creative to support the potential for humans to mitigate environmental conditions.

Much of what is described in the second section on the impacts of LQE would have occurred whether it was caused by climate change or human predation.

The quotes and examples referring to indigenous peoples should be removed (page 3, line 43; page 8, line 40; page 11, line 14). The first paragraph suggests that the concept of “survival of the fittest” justifies destructive colonial actions. Given the positionality of Western research relative to indigenous peoples in colonial contexts, we should be thoughtful about using their words and experiences in our argumentation.

Much of the critique of modern societies, such as “the will to dominate” (page 15, line 52) and supremacy over nature, is related to Western ideologies rather than a global trait of humans. This discussion itself could further support the argument that human beliefs and actions vary. However, as it stands, it could be used to support the fixed nature of humans.

There is an argument that group-level selection may occur within humans when we think of solutions related to mitigating the tragedy of the commons. Laws, policies, and cultural norms become a means of constraining individual behavior to benefit the group. Thus, it could support a way for humans to reduce their environmental impact and become more a part of nature. One thing that humans do that stands out is that we excel at creating a great deal of intra-generational phenotypic variability for evolutionary processes to play around with. To constrain our views on human potential as fixed and hard-wired seems like a limiting belief we should avoid.

---

## [Reviewer Report]

This sort of philosophical musing about the long-term meaning or implications of events and processes in the past does not resolve issues raised in the text. There are no empirical facts, no testing of hypotheses, no falsifying of hypotheses – and therefore no scientific support for either of the polarized positions implied in the manuscript – “anthropomorphic supremacy” (evolution of inevitable human destructiveness) or something else (to be aspired to, such as changing human views towards restoring/conserving ecosystems). Authoritarian religions and hyper-capitalism direct humans to exploit their supremacy, which prehistoric Homo sapiens also might have been in the process of accepting, especially when globally expanding their range.

---

## [Editor Report]

Both reviewers raised concerns with the manuscript. Reviewer 1 finds the two parts of the paper contradictory and suggests they be better integrated, while Reviewer 2 notes that you provide limited evidence to support either position. Please pay careful attention to Reviewer 1’s suggestions for reframing the manuscript to resolve this contradiction. To address Reviewer 2’s commentary, the argument would be strengthened by incorporating additional empirical support.

---

## [Editor Report]

Thank you for addressing the reviews in your revised manuscript. I am happy to accept the manuscript for publication.